# The Effects of Vision-Deprived Progressive Resistance Training on One-Repetition Maximum Bench Press Performance: An Exploratory Study

Ali Boolani [1,2,3,*], Masoud Moghaddam [4], Daniel Fuller [5], Sumona Mondal [5], Shantanu Sur [2], Rebecca Martin [6], Ahmed Kadry [1,7], Ahmed Ali Torad [1,7], Mostafa Ali Elwan [1,8] and Rumit Singh Kakar [9]

1   Department of Physical Therapy, Clarkson University, Potsdam, NY 13699, USA; ahmed_tabia@pt.kfs.edu.eg (A.K.); ahmed_alimohamed@pt.kfs.edu.eg (A.A.T.); mostafa.ali@pt.bsu.edu.eg (M.A.E.)
2   Department of Biology, Clarkson University, Potsdam, NY 13699, USA; ssur@clarkson.edu
3   Honors Program, Clarkson University, Potsdam, NY 13699, USA
4   Health and Human Performance, Salisbury University, Salisbury, MD 21801, USA; mxmoghaddam@salisbury.edu
5   Department of Mathematics, Clarkson University, Potsdam, NY 13699, USA; fullerdt@clarkson.edu (D.F.); smondal@clarkson.edu (S.M.)
6   Department of Physical Therapy, Hanover College, Hanover, IN 47243, USA; martinr@hanover.edu
7   Department of Physical Therapy, Kafrelsheik University, Kafr El Sheik 6860404, Egypt
8   Department of Physical Therapy, Beni-Suef University, Beni-Suef 2722165, Egypt
9   Human Movement Science Department, School of Health Science, Oakland University, Rochester, MI 48309, USA; kakar@oakland.edu
*   Correspondence: aboolani@clarkson.edu

**Abstract:** The objective of this study was to determine whether vision-occluded progressive resistance training would increase upper-extremity movement performance using the one-repetition maximum (1-RM) bench press. Participants ($n = 57$) were recruited from a historically black college and university (HBCU), cross-matched by sex, age ($\pm 1$ year), 1-RM ($\pm 2.27$ kg), 1-RM/weight ($\pm 0.1$), and 1-RM/lean mass ratio ($\pm 0.1$), and randomly assigned to either the experimental group (vision occluded) or the control group. Participants performed resistance training for 6 weeks prior to beginning the study, and 1-RM was assessed the week prior to the beginning of the study. Weight and body composition were measured using a BOD POD. Of the 57 participants who started the study, 34 completed the study (Experimental = 16, Control = 18) and were reassessed the week after completing the 6-week-long training protocol. Using a combination of Mann–Whitney U and Wilcoxon signed-rank tests, we found that when accounting for changes in lean muscle mass, individuals who trained with their vision occluded reported significantly greater improvements in 1-RM strength compared to those who did not ($p < 0.05$). The findings from our study suggest that vision-occluded progressive resistance training increases upper-extremity performance when assessed using the bench press. These findings may have significant practical implications in both sports and rehabilitation, as these techniques may be used to enhance performance in athletes and/or improve rehabilitation effectiveness.

**Keywords:** vision occlusion; bench press; upper extremity; resistance training

## 1. Introduction

While inactive adults experience a 3–8% loss in muscle mass per decade, accompanied by metabolic-rate reduction and fat accumulation, adults who resistance train improve physical performance, movement control, walking speed, functional independence, cognitive abilities, and self-esteem [1]. Resistance training may also enhance cardiovascular health and the prevention and management of type 2 diabetes as well as promote bone

development, and it has been reported to reduce back pain, ease discomfort associated with arthritis and fibromyalgia, and reverse aging-specific factors in skeletal muscles [1]. Resistance training can be performed using machine weights or free weights.

Free-weight resistance training often involves an open kinetic chain. During the performance of exercise, it is vital to continuously update the body and free weight's locations in space relative to each other and various other frames of reference (i.e., location of other equipment used in the exercise). This updating process involves processing and integrating information from various sensory modalities, including proprioception (the sense of the position and movement of our body in space), vestibular (the sense of spatial orientation), and vision [2]. While vision constitutes an important component in performing certain athletic movements such as ballet, dance, and soccer [3–6], for tasks that require high proprioceptor use such as judo or repetitive movement such as triathlon, there is much less dependency on vision with practice [7–10]. Recently, a study in unskilled tennis players reported that total vision occlusion resulted in enhanced forehand-drive performance when compared to the control group [11]. However, little is known about the role played by vision in free-weight resistance training.

We are aware of one study that examined vision-occluded power output in trained (>2 days/week) and untrained individuals [10]. The results from the study suggest that while untrained individuals reported a 11.4% decline in power output when vision was occluded, trained individuals reported no significant change in power output [10]. Power output for this study was calculated during the concentric movement of the leg press, by measuring the distance and time a leg-press footplate was displaced, when lifting 60% of a one-repetition maximum as quickly as possible [10]. However, that study measured a single day of vision-occluded power output and did not have participants performing resistance training with vision occlusion. Based on the previously published literature, vision plays a critical role in resistance training by providing a frame of reference for the weight being lifted [12], especially in novices; however, this dependency on vision in performing the movement decreases over time [7–9].

Vision loss has been shown to increase proprioceptor responses in lower extremities [13,14], and this increase is credited to the body's attempt to improve stability and balance [15,16]. The results from a study on 30 college students suggest that when training on rapid aimed limb movements (wrist rotations) without vision, the movement becomes more accurate and precise with an increase in the number of trials [17]. Taken together, these findings suggest that when vision is occluded, repetitive training will result in increased proprioceptor response, with eventual improvement in movement accuracy and increased strength in the area that is being trained [14]. Due to the higher risk of performing vision-occluded resistance training for lower extremities, we focused on upper-extremity resistance training only in this study, as it allowed the researchers to safely conduct the training protocol. Since the bench press provides an accurate measure of upper-body exercise load [18], this study used the bench press to assess changes in upper-extremity strength. Specifically, this study sought to determine whether vision-occluded resistance training would increase the one-repetition max (1-RM) on the bench press. The authors hypothesized that participants completing the vision-occluded resistance training protocol would have significantly greater increases in 1-RM strength than those in the control group.

## 2. Methods

### 2.1. Design

Using a randomized, controlled, and cross-matched design, the participants were assigned to either the experimental group (vision-occluded resistance training) or the control (resistance training) group. A combination of sex, age ($\pm$1 year), 1-RM ($\pm$2.27 kg), 1-RM/body weight (1-RM/BW) ($\pm$0.1), and 1-RM/lean mass (1-RM/LM) ratio ($\pm$0.1) were used to randomly assign the participants to either the experimental or control group.

### 2.2. Participants

Due to the exploratory nature of this study and the limited literature in the field, an a priori sample size was not calculated. Instead, participants were recruited in the fall and spring until a minimum of 15 had completed the study in both groups.

Participants (*n* = 57, males = 43, females = 14) were recruited from weight-lifting classes at a historically black college and university (HBCU) located in a southern state in the United States. Approval for this study was granted by the University Institutional Review Board (IRB) (approval# HS2011-2692, approved date 30 July 2011), and all participants read and signed the approved consent forms. Participants were administered a health-history questionnaire, which included a survey to determine exposure to weight training.

Participants were eliminated if they had upper-extremity surgery within the past 6 months, had >1 year of weight training experience, <6 months of weight training experience, were a collegiate student athlete, or had consumed performance-enhancing substances within the past 6 months. Out of 152 students recruited, 80 volunteered to participate in the study, among which 57 qualified to participate in the study. Prior to data collection, participants engaged in 6 weeks of resistance training twice per week, in order to become familiar with resistance training (acclimatization phase). During the acclimatization phase of the study, participants performed barbell bench press, lateral pull-down, standing shoulder press, overhead triceps extension, and biceps curl for upper-extremity exercises twice a week during class. Participants were asked to perform 3 sets of each exercise and were instructed that if they could perform 15 repetitions of a certain exercise then they were to increase the weight that they lifted; however, if they were unable to perform 8 repetitions, they were asked to decrease the weight that they were lifting. Participants were instructed to rest as long as they felt they needed to between sets during the first 2 weeks, then for the second two weeks (weeks 3–4) rest time was set at 2.5–3 min, and for the final 2 weeks (weeks 5–6) rest time was reduced to 1.5–2 min between sets.

### 2.3. Protocol

The present study followed a pre–post experimental design. Pre- and post-testing included the anthropometric assessments namely, height, weight, body composition, and 1-RM (i.e., bench press), separated by 7 weeks to allow for the 6-week intervention. The pre-test was completed on the Saturday before the beginning of the intervention, and the post-test was completed on the Saturday after the completion of the intervention. To account for diurnal variations, participants were scheduled for post-testing within 30 min of the time that their pre-test was scheduled (e.g., if pre-test was scheduled at 11 a.m., post-test was scheduled between 10:30 a.m. and 11:30 a.m.). For pre-testing, participants were scheduled for their pre-test date and time on the last day of their acclimatization training. Participants were asked to refrain from consuming coffee and performing exercise 24 h prior to the day of testing and to have their usual night's sleep ($\pm 2$ h of their self-reported usual night's sleep). On the day of testing, participants completed a survey to determine whether they had adhered to the pre-testing day instructions. All participants reported adhering to the pre-testing day instructions. Height and body weight of the participants were measured using a stadiometer and a scale (both without shoes), respectively, and Body Mass Index (BMI) was calculated by dividing the weight in kilograms (kg) by the height in meters squared ($m^2$). The participants' body composition was analyzed using the BOD POD® Body Composition Tracking System, which calculates fat mass and fat-free mass based on Air Displacement Plethysmography (ADP) technology. Afterwards, participants performed the 1-RM bench press on Cybex® Olympic Bench Press with Olympic weightlifting bars (20.45 kg). During the 1-RM test, there was a 5-min rest period between the attempts to allow adequate recovery [19]. If a participant was able to perform 1 repetition, depending on the ease that the participant completed the trial, the lifted weight was increased by a 2.27–4.45 kg increment until failure [19]. The assessment ended when 1 repetition could not be successfully completed. 1-RM bench press has been shown to be an accurate load

predictor for other upper body exercises [18], and, thereby, is an accurate measure of upper extremity strength.

After the pre-testing, the participants were cross-matched by 1-RM, 1-RM/BW, and 1-RM/LM and randomly assigned to either the vision-occluded group or control group. All participants were instructed to refrain from consuming any supplements except for multivitamins. The vision-occluded group consisted of 27 individuals, while the control group consisted of 30 participants; however, 16 participants in the experimental group and 18 participants in the control group completed the entire study. Both the vision-occluded and control groups followed a periodized strength-training protocol [17], including barbell bench press, lateral pull-down, standing shoulder press, overhead triceps extension, and biceps curl. During the first 2 weeks (weeks 1–2) of the training protocol, the participants performed 3 sets of 12–15 repetitions for each exercise. They progressively increased the load and decreased the repetitions to 8–12 times in the next 2 weeks (weeks 3–4), then reduced the repetitions to 6–8 during the last 2 weeks (weeks 5–6). The load progression was determined in such a way that if the participants could perform the higher number of the repetition range, they were allowed to increase the load [17]. If they could not perform the lower number of the repetition range, they were asked to decrease the load. For example, if a participant could perform 15 repetitions on a given exercise in week 1, they were asked to increase the weight; however, if the participant was unable to perform 12 repetitions, they were asked to decrease the weight. The rest periods between the sets were approximately of 1.5–2 min to allow adequate recovery [17]. Prior to the beginning of each exercise, individuals who were blindfolded were asked a minimum of 3 times how many fingers their workout partner was holding up to confirm that the participant's vision was completely occluded. Their lifting partner guided and placed their hands on the weights they were lifting. Participants were allowed to remove their blindfolds between sets if they wanted to, however, vision occlusion was confirmed again prior to the next set. Participants were aware of the weight that they were lifting, as they were allowed to self-select the weight that they were using prior to the beginning of the exercise and/or set.

The participants in the experimental group were deprived of vision through blindfolding. Spotters were employed for both groups during the testing sessions and training protocol to ensure adequate safety. The training protocol was performed 2 times a week for 6 weeks. During the intervention, the participants were requested to not engage in upper-extremity strength training outside the study. If a participant missed 1 session, they were eliminated from the study. After the 6-week intervention, the participants were asked about their supplement use, and the individuals who admitted using supplements, other than multivitamins, were excluded from the study. After the 6-week resistance-training protocol, the participants completed the post-testing session (i.e., anthropometric assessment and 1-RM bench press) following the same process as the pre-testing.

### 2.4. Statistical Analysis
#### 2.4.1. Preliminary Analysis

All data were exported to SPSS® Statistics 26.0 (IBM Corp., Armonk, NY, USA) for analysis. Lean mass was calculated using the body composition results from the BOD POD®. As short-term strength-training programs have been shown to increase lean mass and decrease body fat in both men and women [20], the authors of this study used various techniques to account for changes in lean mass and body fat. A 1-RM to body weight ratio (1-RM/BW) was calculated by dividing the pre- and post-test 1-RM by the pre- and post-test body weight (BW), respectively. Further, a 1-RM to lean mass (LM) ratio was calculated by dividing the pre- and post-test 1-RM by the pre- and post-test LM. Percentage changes in 1-RM, BW, LM, 1-RM/BW, and 1-RM/LM were calculated by subtracting the post-test measures from the pre-test measures, dividing the difference by the pre-test measure, and multiplying by 100 ((post-pre)/pre × 100). Changes in 1-RM were divided by changes in lean mass, and further percentage changes in 1-RM were divided by percentage changes in BW and LM. Variables were tested for normality using a combination

of Shapiro–Wilks test ($p > 0.05$) and histograms. If a variable was not normal, normalization techniques were applied, and if data could not be normalized, non-parametric analyses were used. Pre-testing data were compared between groups using a combination of *t*-tests and Mann–Whitney U tests, and no significant differences were noted between groups (Table 1).

### 2.4.2. Primary Analysis

Due to the non-parametric nature of this data, Mann–Whitney U test and a Wilcoxon signed-rank test were used to compare the difference between pre- and post-test data. The primary interests were the presence of a statistically significant difference for pre and post-test measures ($\Delta$) between groups. The Mann–Whitney U test was used to identify differences between the groups for 1-RM $\Delta$, 1-RM $\Delta$ (%), 1-RM/BW $\Delta$, 1-RM/BW $\Delta$ (%), 1-RM/LM $\Delta$, 1-RM/LM $\Delta$, 1-RM $\Delta$ (%)/BW $\Delta$(%), and 1-RM $\Delta$ (%)/LM $\Delta$ (%). The Wilcoxon signed-rank test was used to identify differences for pre- and post-training measures within groups.

### 2.4.3. Post Hoc Analyses

Effect sizes were calculated using the following formula $\eta^2 = Z^2/(N)$ [21]. Due to the large number of analyses and the exploratory nature of this study, we used the Benjamini-Hochberg False Discovery Rate (FDR) to correct for Type I errors and considered 20% FDR as an acceptable error [22].

**Table 1.** Participant characteristics and associated statistical results. The sample size is represented by 'n', while the statistics used include the W statistic, the F statistic, the X$^2$ statistic, and the ∪ statistic. Statistical testing was performed in three separate manners: for the experimental and control groups before and after testing (the "Experimental" and "Control" columns, respectively), for both groups together before and after testing (the "Both groups" column), and between groups (the "Between groups" column). Statistical significance was thresholded at $p < 0.05$, and the effect size is reported as η$^2$. Abbreviations: RTE = resistance training experience, 1-RM = one-repetition maximum.

| | Experimental (n = 16) | | | | | Control (n = 18) | | | | | Between Groups | | Both Groups (n = 34) | | |
|---|---|---|---|---|---|---|---|---|---|---|---|---|---|---|---|
| Variable | Descriptive Stats | | Pre–Post Differences | | | Descriptive Stats | | Pre–Post Differences | | | F/X²/∪ | p | Pre–Post Differences | | |
| Males:Females | 11:5 | | | | | 13:5 | | | | | 0.049 | 0.824 | | | |
| RTE (months, mean ± SD) | 8.3 ± 0.7 | | | | | 8.1 ± 0.8 | | | | | −0.771 | 0.780 | | | |
| Variable | Range | Median | W | p | η² | Range | Median | W | p | η² | F/X²/∪ | p | W | p | η² |
| Height (cm) | 154.94, 190.50 | 176.53 | | | | 154.94, 186.69 | 175.26 | | | | 118.50 | 0.384 | | | |
| Pre-testing weight (kg) | 61.24, 137.17 | 79.51 | 72.00 | 0.836 | 0.003 | 58.97, 126.10 | 79.61 | 77.50 | 0.728 | 0.007 | 142.00 | 0.959 | 291.00 | 0.912 | <0.001 |
| Post-testing weight (kg) | 19.94, 42.31 | 25.42 | | | | 19.66, 36.68 | 26.34 | | | | 140.00 | 0.905 | | | |
| Pre-testing BMI (kg/m²) | 19.94, 42.32 | 25.42 | 70.00 | 0.918 | 0.001 | 19.65, 36.68 | 26.34 | 78.00 | 0.744 | 0.006 | 142.00 | 0.959 | 285.00 | 0.831 | 0.001 |
| Post-testing BMI (kg/m²) | 19.80, 42.29 | 25.37 | | | | 19.23, 37.60 | 26.39 | | | | 144.00 | 0.999 | | | |
| Pre-testing body fat percentage (%) | 5.2, 43.5 | 21.5 | 35.50 | 0.286 | 0.074 | 5.6, 37.4 | 19.95 | 36.00 | 0.055 | 0.209 | 139.00 | 0.878 | 142.00 | 0.038 | 0.119 |
| Post-testing body fat percentage (%) | 5.60, 43.40 | 20.65 | | | | 6.30, 36.80 | 18.70 | | | | 137.00 | 0.825 | | | |
| Pre-testing lean mass (kg) | 42.13, 94.10 | 66.16 | 97.00 | 0.134 | 0.146 | 46.45, 84.99 | 67.02 | 106.00 | 0.372 | 0.046 | 134.00 | 0.746 | 397.00 | 0.089 | 0.080 |
| Post-testing lean mass (kg) | 45.88, 87.73 | 67.04 | | | | 47.59, 94.50 | 68.38 | | | | 136.00 | 0.798 | | | |
| Pre-testing 1-RM (kg) | 24.95, 133.81 | 78.24 | 101.50 | 0.002 | 0.619 | 20.41, 117.93 | 75.98 | 122.00 | 0.005 | 0.448 | 144.00 | 0.999 | 430.00 | <.001 | 0.462 |
| Post-testing 1-RM (kg) | 29.48, 138.35 | 82.78 | | | | 22.68, 124.74 | 82.78 | | | | 146.00 | 0.959 | | | |
| Pre-testing 1-RM/weight ratio | 0.24, 1.24 | 0.94 | 128.00 | 0.002 | 0.623 | 0.27, 1.54 | 0.93 | 149.00 | 0.006 | 0.437 | 118.00 | 0.737 | 538.00 | <.001 | 0.469 |
| Post-testing 1-RM/weight ratio | 0.39, 1.27 | 0.98 | | | | 0.31, 1.57 | 1.02 | | | | 123.00 | 0.882 | | | |
| Pre-testing 1-RM/lean mass ratio | 0.43, 1.15 | 0.43 | 112.00 | 0.023 | 0.345 | 0.38, 1.83 | 0.39 | 134.00 | 0.035 | 0.254 | 125.00 | 0.941 | 485.00 | 0.001 | 0.286 |
| Post-testing 1-RM/lean mass ratio | 0.64, 1.62 | 1.19 | | | | 0.42, 1.77 | 1.27 | | | | 137.00 | 0.737 | | | |

### 3. Results

*3.1. Participants*

Out of the 57 participants who completed the initial testing, 53 completed some or most of the study, and 34 completed the full study. Of the 23 whose data were not included in the analysis, 5 were eliminated due to missing training sessions, 14 were eliminated due to missing the day when body composition was measured, and 4 reported consuming supplements during the duration of the study (see Figure 1). Of the 34 participants who completed the study, 16 were in the experimental condition (males = 11, females = 5) and 18 were in the control condition (males = 13, females = 5). There was no significant difference ($p > 0.05$) in baseline measurements for 1-RM, height, weight, BMI, body composition, and number of months of training experience. Additional information on participant demographics can be found in Table 1.

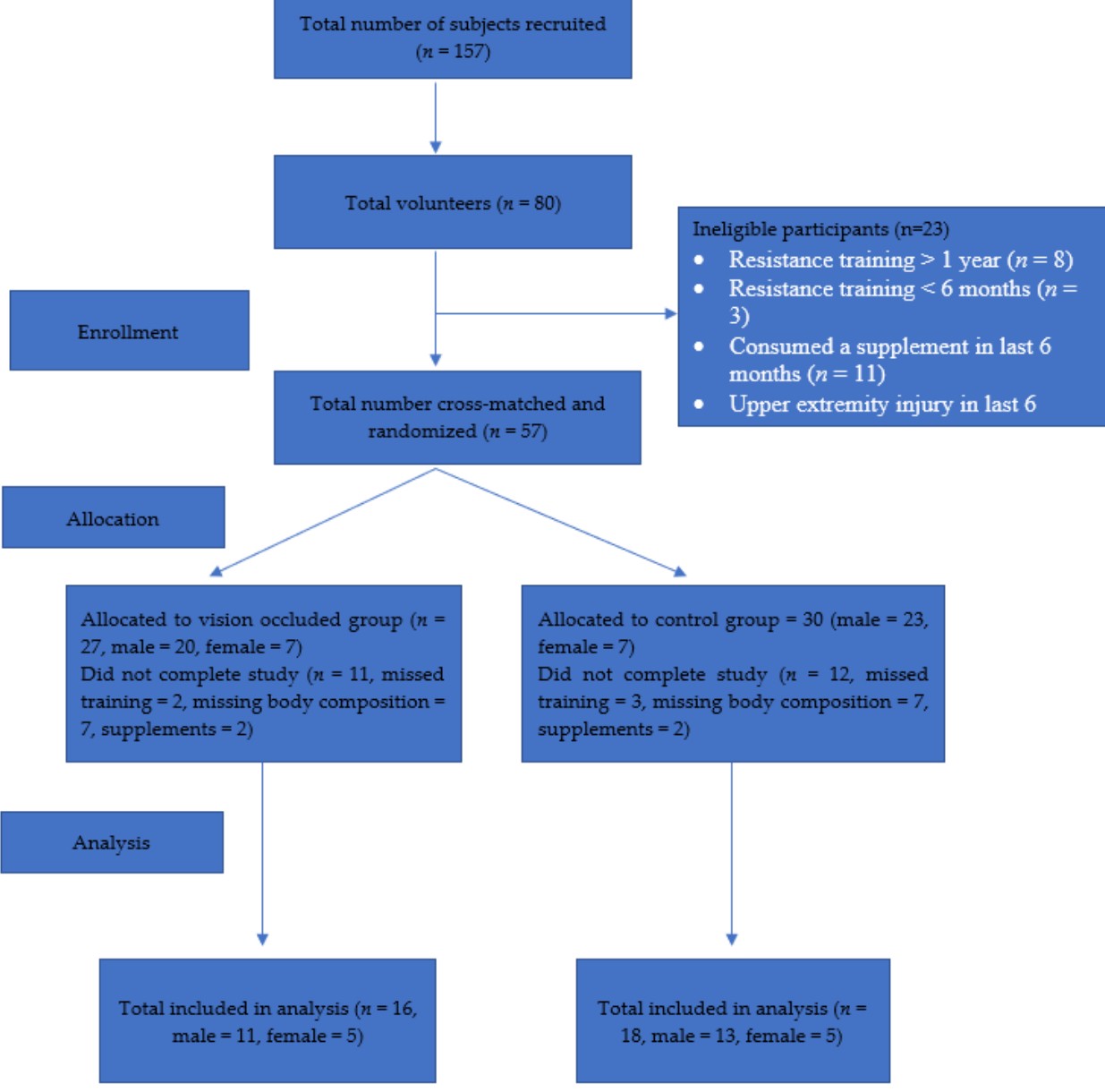

**Figure 1.** CONSORT flow diagram.

### 3.2. Pre–Post Differences between Groups

In Table 1, the Mann–Whitney U test results suggest that when comparing differences between groups, there were no significant differences ($p > 0.05$) between changes in 1-RM, 1-RM/BW ratio, 1-RM/LM ratio, BW, LM, Body Mass Index (BMI), or body fat percentage. When comparing the percentage change in 1-RM, 1-RM/BW ratio, 1-RM/LM ratio, BW, LM, Body Mass Index (BMI), or body fat percentage, there were no significant differences between groups ($p > 0.05$). However, when using the Mann–Whitney U test to examine the ratio between changes in 1-RM and changes in lean mass (Figure 2) and percentage changes in 1-RM and percent changes in LM (Figure 3), the results indicate that individuals who were in the experimental group showed significantly greater increases in 1-RM/LM strength ratio compared to the control group (Figure 2).

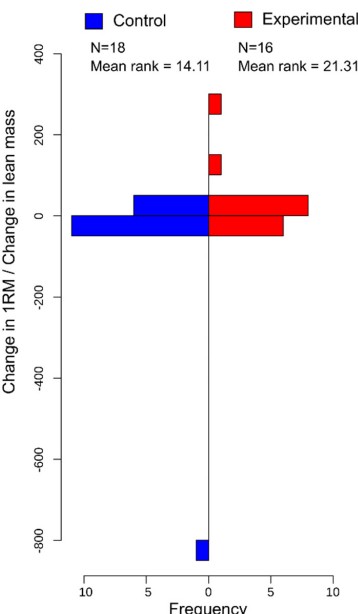

**Figure 2.** Histogram displaying frequencies of the ratio between change in 1-RM and change in lean mass between the control and experimental group. Ranking was performed for use in the Mann–Whitney U test.

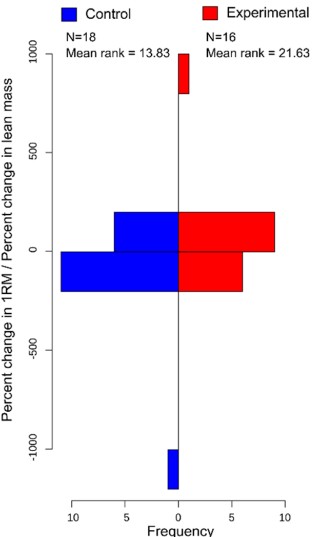

**Figure 3.** Histogram displaying frequencies of the ratio between percentage change in 1-RM and percentage change in lean mass between the control and experimental group. Ranking was performed for use in the Mann–Whitney U test.

### 3.3. Pre–Post Differences for Both Groups

Analysis yielded significant improvement ($p < 0.05$) in body composition (body fat percentage), 1-RM, 1-RM/BW, and 1-RM/LM within groups (Table 1). Overall, all participants had a significant decrease in body fat percentage (W = 142.00, $p = 0.038$, Mean $\pm$ SD = 0.71 $\pm$ 2.78%, median = 0.65%, $\eta^2 = 0.123$), improvements in 1-RM (W = 430.00, $p < 0.001$, Mean $\pm$ SD = 5.34 $\pm$ 5.80 kg, median = 4.54 kg, $\eta^2 = 0.476$), 1-RM/BW ratio (W = 538, $p < 0.001$, Mean $\pm$ SD = 0.06 $\pm$ 0.7, median = 0.07, $\eta^2 = 0.476$), and 1-RM/LM ratio (W = 485, $p = 0.001$, Mean $\pm$ SD = 0.07 $\pm$ 0.11, median = 0.09, $\eta^2 = 0.483$).

### 3.4. Pre–Post Differences for Experimental Group Only

When comparing pre–post differences in the experimental group only, our findings suggest significant improvement in 1-RM performance (W = 101.50, $p = 0.002$, Mean $\pm$ SD = 5.24 $\pm$ 5.41 kg, median = 4.54 kg, $\eta^2 = 0.638$), 1-RM/BW (W = 128.00, $p = 0.002$, Mean $\pm$ SD = 0.06 $\pm$ 0.06, median = 0.07, $\eta^2 = 0.642$) and 1-RM/LM (W = 112.00, $p = 0.023$, Mean $\pm$ SD = 0.07 $\pm$ 0.10, median = 0.07, $\eta^2 = 0.345$).

### 3.5. Pre–Post Differences for Control Group Only

When comparing pre–post differences in the control group only, our findings suggest significant improvements in 1-RM performance (W = 122.00, $p = 0.005$, Mean $\pm$ SD = 5.41 $\pm$ 6.28 kg, median = 4.54 kg, $\eta^2 = 0.462$), 1-RM/BW (W = 149.00, $p = 0.006$, Mean $\pm$ SD = 0.07 $\pm$ 0.08, median = 0.06, $\eta^2 = 0.450$), 1-RM/LM (W= 134.00, $p = 0.035$, Mean $\pm$ SD = 0.07 $\pm$ 0.12, median = 0.09, $\eta^2 = 0.262$). While not significant, there was a decrease in body fat in the control group approaching significance ($p = 0.055$, Mean $\pm$ SD = 0.71 $\pm$ 2.81%, median = 0.75%).

### 3.6. Post Hoc

For the post-test effect size for the Wilcoxon signed-rank test, pre–post changes in 1-RM for the vision-occluded group only was 0.619, while the change in the 1-RM/BW ratio was 0.623 and the 1-RM/LM ratio was 0.345. When examining pre–post changes in 1-RM for the control group only, we report an effect size of 0.448, while the change in 1-RM/BW ratio has an effect size of 0.437, and 1-RM/LM ratio has an effect size of 0.254. When calculation for the effect sizes for the Mann–Whitney U test comparing pre–post differences between groups for $\Delta$1-RM/$\Delta$LM (Figure 2) has an effect size of 0.134, %$\Delta$1-RM/%$\Delta$LM (Figure 3) has an effect size of 0.157 (Table 2).

**Table 2.** Pre–post differences (range, median).

| Variable | Experimental (*n* = 16) | | Control (*n* = 18) | | ∪ | *p*-Value | $\eta^2$ |
|---|---|---|---|---|---|---|---|
| | Range | Median | Range | Median | | | |
| Weight $\Delta$ (kg) | −2.00, 2.27 | −0.18 | −2.81, 4.72 | −0.41 | 291.00 | 0.912 | 0.008 |
| Weight $\Delta$ (%) | −1.48, 3.03 | −0.18 | −3.55, 5.82 | −0.48 | 121.00 | 0.443 | 0.018 |
| BMI $\Delta$ | −0.55, 0.76 | −0.06 | −0.83, 1.54 | −0.13 | 127.00 | 0.574 | 0.010 |
| BMI $\Delta$ (%) | −1.45, 3.12 | −0.18 | −0.48, −3.44 | −0.48 | 121.00 | 0.443 | 0.018 |
| Body fat % $\Delta$ | −5.90, 5.60 | −0.40 | −5.70, 8.10 | −0.75 | 132.00 | 0.695 | 0.003 |
| Body fat % $\Delta$ (%) | −18.91, 17.83 | −1.37 | −24.39, 31.89 | −3.86 | 134.00 | 0.730 | 0.010 |
| Lean Mass $\Delta$ (kg) | −8.94, 5.96 | 0.30 | −5.00, 9.51 | 0.38 | 127.00 | 0.574 | 0.003 |
| Lean Mass $\Delta$ (%) | −9.50, 8.90 | 0.44 | −8.75, 11.19 | 0.58 | 134.00 | 0.746 | 0.002 |
| 1-RM $\Delta$ (kg) | −4.54, 18.14 | 4.54 | −6.80, 15.88 | 4.54 | 151.50 | 0.798 | 0.002 |
| 1-RM $\Delta$ (%) | −4.17, 61.54 | 5.81 | −8.70, 38.46 | 6.23 | 150.50 | 0.825 | 0.002 |
| 1-RM/body weight $\Delta$ | −0.04, 0.18 | 0.07 | −0.09, 0.20 | 0.06 | 152.00 | 0.798 | 0.004 |
| 1-RM/lean mass $\Delta$ | −0.14, 0.26 | 0.07 | −0.17, 0.27 | 0.09 | 149.00 | 0.878 | 0.001 |

**Table 2.** *Cont.*

| Variable | Experimental (*n* = 16) | | Control (*n* = 18) | | ∪ | *p*-Value | $\eta^2$ |
|---|---|---|---|---|---|---|---|
| | **Range** | **Median** | **Range** | **Median** | | | |
| Δ1-RM/ Δ lean mass | −4.49, 254.66 | 1.10 | −806.13, 20.52 | −1.49 | 83.00 | 0.036 | 0.130 |
| 1-RM/weight ratio Δ (%) | −4.74, 61.66 | 6.52 | −10.59, 35.26 | 7.86 | 154.00 | 0.746 | 0.002 |
| 1-RM/lean mass ratio Δ (%) | −10.67, 61.37 | 6.06 | −17.56, 51.74 | 7.29 | 152.00 | 0.798 | 0.004 |
| 1-RM Δ (%)/body weight Δ (%) | −831.31, 28.21 | −3.74 | −38.96, 50.86 | −1.85 | 154.00 | 0.746 | 0.004 |
| 1-RM Δ (%)/lean mass Δ (%) | −6.00, 863.73 | 1.72 | −1184.81, 27.41 | −1.93 | 78.00 | 0.022 | 0.152 |

The post hoc Benjamini-Hochberg FDR was conducted using 54 analyses and an FDR rate of 0.20. All q-values were less than the calculated *p*-values for the primary analyses.

## 4. Discussion

The objective of this exploratory study was to determine whether vision-occluded resistance training improved 1-RM performance on the bench press. The results of this study suggest that both groups had a moderate increase in 1-RM, 1-RM/BW, and 1-RM/LM. However, individuals performing vision-occluded resistance training significantly improved their 1-RM performance on the bench press, when accounting for changes in lean mass. To the best of our knowledge, this is the first study to use vision-occluded resistance training during all resistance-training exercises to improve 1-RM performance.

When examining pre–post differences in individual groups, we find that both groups had significant improvements in their 1-RM, 1-RM/BW, and 1-RM/LM, however, when examining effect sizes, we find that individuals in the vision-occluded group had greater effect sizes when compared to the control groups (0.623 vs. 0.469). These findings suggest that while both groups reported improvements, the vision-occluded group experienced greater improvements in 1-RM performance. These findings were further validated when examining 1-RM differences, while accounting for lean mass. Here we find individuals in the vision-occluded group had small ($\eta^2$ = 0.134 and 0.157) yet significantly greater improvements in 1-RM performance, when compared to the control group. Taken together, the findings of our study suggest that progressive resistance-training programs significantly improve 1-RM performance, however, individuals who perform vision-occluded progressive resistance training see a small, but significantly greater improvement in their 1-RM performance.

To our knowledge there is only one other study (a published dissertation) that examined vision-occluded resistance training. The study examined the effects of vision-occluded resistance training on changes in 1-RM on the bench press [23]. Like our study, the results of that study did not find significant differences in 1-RM, however, based on the findings of our study, we believe that the primary reason for our significant findings is that the prior study did not account for changes in lean mass when measuring changes in 1-RM strength. As previously reported, increased lean mass following a short-term bout of resistance training is associated with changes in muscular strength [20]. Another difference between the two studies is that this study had participants performing all resistance-training exercise with vision occlusion, while the previous study had participants in the experimental group train on the bench press with vision occlusion, while all other exercises were performed with vision [23].

Although this study did not measure changes in proprioceptor responses or movement efficiency, we hypothesize that continuous training with vision occlusion may have enhanced neural activation of proprioceptors [13,14], reduced co-activation of antagonists [24], and optimized activation of synergists and agonists [25,26], compared to those who trained with vision. Although it was not the original intention of the study to control for resistance-training experience, a granular look at the data shows that participants were matched ±1 month of resistance-training experience (i.e., someone with 7 months of resistance training experience was matched with someone with either 6, 7, or 8 months of

experience). Based on how similar our participants were at baseline, these findings suggest that vision-occluded resistance training increased 1-RM strength relative to muscle mass. These findings may be compared to the literature on training in unstable environments, as those environments lead to increase proprioceptor activation, given the lack of reliable input from the visual system resulting in increased isotonic strength in young adults [27].

### 4.1. Practical Applications

While this study did not assess the practical implications of this protocol, the findings of this study may have substantial implications for sports performance and rehabilitation research. Visual occlusion has been used to train perceptual ability in sports such as soccer [28–30]. Paired with the potential strength-gains reported in this study, visual occlusion could be a beneficial tool for training athletes. Moreover, visually impaired individuals have been reported to have reduced strength and balance. Resistance training has been shown to have beneficial effects in rehabilitation and for regaining motor control and strength in different population groups [31–33]. Follow-up studies targeting upper- and lower-extremity strength training with visual occlusion could provide insights into developing training programs for visually impaired para-athletes.

### 4.2. Limitations

This study is not without limitations. A primary limitation of this study is that we did not measure whether changes in bench-press strength occurred due to improvement in motor-movement accuracy or if there were other external factors involved. Secondly, we were reliant on self-reported compliance to our pre- and post-testing instructions as well as self-reported use of supplements. Additionally, approximately 40% of the study population were not included in the final analysis, which may limit the findings of this study. Although, none of the participants dropped out of the study, 5 out of 57 missed one session, 4 out of 57 reported taking supplements during the study, and 14 out of 57 did not report for their body-composition testing. Due to the university schedule, all interventions had to be completed in the 8 weeks after the mid-semester break and before final exams. Body-composition assessments were scheduled the Saturday before final exams began, which may have limited the number of people who reported to the lab. Further, while most exercise studies set a standard for the minimal number of sessions that participants must attend to be included in the study, due to the exploratory nature of this study, the authors chose to eliminate any participants who missed 1 day of the intervention. However, we believe that the results of these findings warrant additional investigation with larger trials to determine the efficacy of using vision-occluded resistance training on improvements in muscular strength. We recommend exploring the factors that may relate to the current findings by using functional Magnetic Resonance Imaging (fMRI) to explore brain activities, electromyography (EMG) and functional Near-Infrared Spectroscopy (fNIRS) to explore brain and muscle activity, and oxygen consumption and isokinetic movements to explore differences in both eye-opened and vision-occluded conditions.

### 5. Conclusions

The objective of this study was to determine the effects of vision-occluded resistance training on 1-RM performance on the bench press. Our findings suggest that short-term resistance training improved muscular strength and body composition in both groups. However, vision-occluded resistance training accounted for a greater improvement in 1-RM on the bench press, when considering the changes in lean mass. The results of this study indicate the need for larger trials to identify the use of vision-occluded resistance training to increase strength in athletes. Despite not being tested in a clinical population, this protocol has the potential to induce a beneficial impact on rehabilitation protocols, as vision occlusion may be used to improve muscular strength in patient populations. Furthermore, the strategy could be implemented in training regimens for athletes and other healthy adults.

**Author Contributions:** IRB submission: A.B.; data collection: A.B; data curation: A.K., A.A.T., M.A.E. and A.B.; data visualization: A.B., D.F., M.M. and S.M.; data analysis: A.B., D.F., S.M., A.K. and A.A.T.; initial draft: M.M., D.F., A.A.T., S.M., S.S., A.K., R.S.K., M.A.E., A.B. and R.M.; final draft review: M.M., D.F., A.A.T., S.M., S.S., A.K., R.S.K., M.A.E., A.B. and R.M. All authors have read and agreed to the published version of the manuscript.

**Funding:** This project received no external funding.

**Institutional Review Board Statement:** The study was conducted in accordance with the Declaration of Helsinki, and approved by the Institutional Review Board (or Ethics Committee) of Tennessee State University (protocol code HS2011-2692 and date of approval was 30 July 2011).

**Informed Consent Statement:** Informed consent was obtained from all subjects involved in the study.

**Data Availability Statement:** Data may be obtained by contacting the corresponding author A.B.

**Acknowledgments:** We would like to thank the weight training instructors at Tennessee State University for allowing us to collect data in their classes. We would also like to thank Essam Hamido and Timothy Jones for helping with data collection.

**Conflicts of Interest:** The authors declare no conflict of interest.

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
