# Peer review of "The Effects of Vision-Deprived Progressive Resistance Training on One-Repetition Maximum Bench Press Performance: An Exploratory Study"

_2411-5150, 2022_

Round 1

Reviewer 1 Report

Dear reviewers,

Thanks you for this interesting paper. I consider you have improved significantly the article. Some final concerns about it:

  • Introduction: The framework should be improved. I would like to star with a little paragraph about resistance training (strength concept, importance to improve strength to health/performance...), if there are some oclusion studies with other variables (like equilibrium, endurance...) and finally, the different mechanism that influence in propioception and vestibular system. On the other hand,  The hypothesis from the first paragraph should be removed or link to the final hypothesis
  • Protocol: it has been improved and its well described and based of scientific evidence. 
  • Statistical analysis: I willl include this section in method not apart from this.
  • Results: Due to the small size sample, Have you considered the possibility of using Cohen's effect size? If not, please explain the formula of Tomczak study formula
  • References: include doi from all references.

Author Response

Dear reviewer,

We would like to thank the reviewer for taking the time to review our manuscript. Please see our responses to the reviewers concerns below in red.

Thanks you for this interesting paper. I consider you have improved significantly the article. Some final concerns about it:

Thank you for the kind words.

  • Introduction: The framework should be improved. I would like to star with a little paragraph about resistance training (strength concept, importance to improve strength to health/performance...), if there are some oclusion studies with other variables (like equilibrium, endurance...) and finally, the different mechanism that influence in propioception and vestibular system. On the other hand,  The hypothesis from the first paragraph should be removed or link to the final hypothesis

We appreciate the reviewer’s feedback. Based on this reviewer’s feedback we have written a paragraph on the benefits of resistance training. Further have the role of vision occlusion training on table tennis forehand drive performance.

We have now taken out “Based on previously published literature, we hypothesize” as this sentence is supported by previously published evidence.

  • Protocol: it has been improved and its well described and based of scientific evidence. 

Thank you

  • Statistical analysis: I willl include this section in method not apart from this.

We agree with this reviewer that the statistical analysis section should be in the methods section. We have now moved it to the methods section.

  • Results: Due to the small size sample, Have you considered the possibility of using Cohen's effect size? If not, please explain the formula of Tomczak study formula

We appreciate the reviewer’s comment. The Tomczak study provides a formula for calculating Cohen’s effect size for non-parametric tests.

  • References: include doi from all references.

We have added doi numbers for references where we could find the DOIs. However, if the manuscript is accepted we will support MDPI in finding the DOIs.

Reviewer 2 Report

Tables are strange for me but I understand them

Author Response

Dear reviewer,

We would like to thank the reviewer for taking the time to review our manuscript. Please see our responses to the reviewers concerns below in red.

Tables are strange for me but I understand them

We have now adjusted the tables.

Reviewer 3 Report

General Comments

This paper examined the use of vision occlusion on various strength measures, both absolute and relative. The topic is novel and interesting to both researchers and practitioners alike. The study appears well-designed and executed. However, the presentation of the results, particularly as it relates to tables and figures, is either confusing or incomplete. I have detailed some of these concerns below.

Specific Comments

  1. Abstract: You did a good job of controlling and matching for absolute and relative strength. Did you make an effort to control for total resistance training experience? For example, two individuals may have the same 1-RM strength but different training histories.
  2. Introduction, paragraph 1: It would be helpful to include more information on how the cited previous study measured power output in trained and untrained individuals.
  3. Introduction, paragraph two, second sentence: What type of training and movement?
  4. Methods, paragraph one: How was this study "placebo-controlled?"
  5. Participants section, last sentence: Can you describe what was done in the resistance training program during the familiarization (acclimatization) phase? For example, exercises, sets, repetitions, progression, etc.
  6. Protocol section: More information on blindfolding is needed. What type of blindfold? Did you verify that absolutely no vision was possible? Were participants also "blinded" to weight progressions during the lifting sessions (for example, increases in weight with strength increases)?
  7. Table 1 is very difficult to read and interpret. Consider reorganizing or dividing into multiple tables. Consider also providing the key to the various abbreviations and acronyms (for example, W = ?, n2 = ?, etc.), for readers who may not be familiar with them.
  8. Also, a summary is needed. What do these numbers in table 1 tell us about your results? Tables should "stand alone" and not need reference back to the text for interpretation. Where are the significant differences/changes, etc.?
  9. Table 1: Nowhere in the previous text have you mentioned calculating BMI or its importance to your study.
  10. Results: Why did you choose this specific order to state your results? Section 4.5 should be presented first since this is the purpose of your study (comparing gains for experimental vs. control group). The other changes are not particularly interesting (it is expected that strength training will increase strength).
  11. Results, section 4.5: I think you need to be more specific about "showed significant improvements" in the last sentence. Are you saying that there were greater 1-RM to lean mass ratios, etc., for the experimental group? If so, this would suggest you believe that gains in strength from neural adaptations are more desirable than gains in strength due to hypertrophy, etc.
  12. Results, section 4.5: I assume the last sentence should refer to figure 2, not figure 1?
  13. Figures 2 and 3: Summarize what these figures tell us in the captions. Also, reduce the number of decimal places to facilitate ease of reading.
  14. Table 2: Delta 1-RM/Delta Lean mass; can you verify that the numbers for the experimental and control groups are correct (for example, -806.13)?
  15. Discussion, last sentence of the third paragraph: Consider changing "benefit of vision" to "with vision." According to your results, having vision during lifting is NOT a "benefit."

Author Response

Dear reviewer,

We would like to thank the reviewer for taking the time to review our manuscript. We believe the manuscript is significantly stronger due to the reviewer’s thorough comments. Please see our responses to the reviewers concerns below in red.

This paper examined the use of vision occlusion on various strength measures, both absolute and relative. The topic is novel and interesting to both researchers and practitioners alike. The study appears well-designed and executed. However, the presentation of the results, particularly as it relates to tables and figures, is either confusing or incomplete. I have detailed some of these concerns below.

Specific Comments

  1. Abstract: You did a good job of controlling and matching for absolute and relative strength. Did you make an effort to control for total resistance training experience? For example, two individuals may have the same 1-RM strength but different training histories.

We appreciate this feedback and although it was not our original intention to match participants based on resistance training experience, we did go back to the original data file and find that the subjects were matched ±1 month of resistance training (i.e. someone with 7 months of resistance training experience was matched with someone with 6, 7 or 8 months of resistance training experience). Since this was not one of the original matching criteria of the study we have decided not to report as a primary way to match subjects, we did report this in our discussion section.

  1. Introduction, paragraph 1: It would be helpful to include more information on how the cited previous study measured power output in trained and untrained individuals.

We have now added a sentence regarding the way power output was measured in the study by Killebrew and colleagues.

  1. Introduction, paragraph two, second sentence: What type of training and movement?

We have added information regarding the type of training that was performed.

  1. Methods, paragraph one: How was this study "placebo-controlled?"

We appreciate the reviewer catching this error. We have updated it to no longer state placebo-controlled.

  1. Participants section, last sentence: Can you describe what was done in the resistance training program during the familiarization (acclimatization) phase? For example, exercises, sets, repetitions, progression, etc

We appreciate the reviewer asking for this information as we believe that this information will make the study easier to replicate. Below is what was added to the manuscript

During the acclimatization phase of the study participants performed barbell bench press; lateral pull-down; standing shoulder press; overhead triceps extension; and biceps curl for upper extremity exercises twice a week during class. Participants were asked to perform 3 sets of each exercise and were instructed that if they could perform 15 repetitions of a certain exercise then they were to increase the weight that they lifted however, if they were unable to perform 8 repetitions they were asked to decrease the weight that they were lifting. Participants were instructed to rest as long as they felt they needed to between sets during the first 2 weeks, for the second two weeks (weeks 3-4) rest time was set at 2.5-3 minutes and for the final 2 weeks (weeks 5-6), rest time was reduced to 1.5-2 minutes between sets.

  1. Protocol section: More information on blindfolding is needed. What type of blindfold? Did you verify that absolutely no vision was possible? Were participants also "blinded" to weight progressions during the lifting sessions (for example, increases in weight with strength increases)?

We appreciate the reviewer asking for an explanation of this as we feel that it makes it easier to replicate the study. We have now added this information in our protocol. Below is what we added to the manuscript.

Prior to the beginning of each exercise, individuals who were blindfolded were asked a minimum of 3 times how many fingers their workout partner was holding up to confirm that the participant’s vision was completely occluded. Their lifting partner guided and placed their hands on the weights they were lifting. Participants were allowed to remove their blindfolds between sets if they wanted to however, vision occlusion was confirmed again prior to the next set. Participants were aware of the weight that they were lifting as they were allowed to self-select the weight that they were using prior to the beginning of the exercise and/or set.

  1. Table 1 is very difficult to read and interpret. Consider reorganizing or dividing into multiple tables. Consider also providing the key to the various abbreviations and acronyms (for example, W = ?, n2 = ?, etc.), for readers who may not be familiar with them.

We have now tried to make the table 1 clearer while also providing a legend.

  1. Also, a summary is needed. What do these numbers in table 1 tell us about your results? Tables should "stand alone" and not need reference back to the text for interpretation. Where are the significant differences/changes, etc.?

We have provided p-values in the tables for interpretations for differences/changes.

  1. Table 1: Nowhere in the previous text have you mentioned calculating BMI or its importance to your study.

We have now reported in our protocol section how BMI was calculated.

  1. Results: Why did you choose this specific order to state your results? Section 4.5 should be presented first since this is the purpose of your study (comparing gains for experimental vs. control group). The other changes are not particularly interesting (it is expected that strength training will increase strength).

We appreciate the reviewer’s feedback. We have now changed the order to report the differences between experimental and control group. We felt that we needed to report the other results since the effect sizes η2 for the experimental group was much higher than the η2 the control group. These findings provide further evidence that although both groups reported statistically significant improvements in strength the effect size for one group was larger than the effect size for the other group.

  1. Results, section 4.5: I think you need to be more specific about "showed significant improvements" in the last sentence. Are you saying that there were greater 1-RM to lean mass ratios, etc., for the experimental group? If so, this would suggest you believe that gains in strength from neural adaptations are more desirable than gains in strength due to hypertrophy, etc.

We have now adjusted the sentence to state “significantly greater increases in 1RM/LM strength ratio compared to the control group”

  1. Results, section 4.5: I assume the last sentence should refer to figure 2, not figure 1?

Thank you for catching this error, we have adjusted this to state Figure 2.

  1. Figures 2 and 3: Summarize what these figures tell us in the captions. Also, reduce the number of decimal places to facilitate ease of reading.

We have now summarized the figures in the titles and we have reduced the decimals to 2 places.

  1. Table 2: Delta 1-RM/Delta Lean mass; can you verify that the numbers for the experimental and control groups are correct (for example, -806.13)?

We went back and verified and the numbers are accurate. For example, the person who had a -1184.81 change in delta 1-RM/delta lean mass had a lean mass change of -0.0164% change in LM and a 13.33% change in 1RM.

  1. Discussion, last sentence of the third paragraph: Consider changing "benefit of vision" to "with vision." According to your results, having vision during lifting is NOT a "benefit."

We have now corrected this to state “with vision”

Round 2

Reviewer 3 Report

Thank you for addressing my comments and concerns.

This manuscript is a resubmission of an earlier submission. The following is a list of the peer review reports and author responses from that submission.

Round 1

Reviewer 1 Report

I consider it to be a very necessary and interesting research topic. However, it is very difficult to understand the manuscript. The results and the methodology are very confusing. The authors should rewrite them.
Other topics:
- How was the sample size calculated?
- Point 4.4 at p <0.005, 0 is missing.
- In the participants, since there are more men than women, can it influence the results?
- In the discussion it is mainly compared with articles from before 2010. I recommend that these publications be updated.

Reviewer 2 Report

Dear authors.

Thanks you so much for this interesting paper. Some concerns:

  • Introduction: it should be improved. From my point of view it could be interested to included more references about strength training and protocolos to train (Why you use your training protocol?). Hypothesis should be included and de main purpose of the study
  • Method: Have this study approval by some ethical committee o similar? 
  • Results/Discussion:  In the results apart I would like to see more clarify some aspects. For instance, you say that pre-post RM is higher in experimental group, but in discussion you say that the size effect was higher in the control group. It should be clarifies. On the other hand I suggest the authors to improve discussion section with more reference (similar comment for the introduction section)
  • Limitations: Why a 40% of the sample didn't finished the study? It should be taking into account because the protocol probably had some problems to be ensure the possibility of complete the study

Good luck!